# Evidence of Critical Dynamics in Movements of Bees inside a Hive

**DOI:** 10.3390/e24121840

**Published:** 2022-12-17

**Authors:** Ivan Shpurov, Tom Froese

**Affiliations:** Embodied Cognitive Science Unit, Okinawa Institute of Science and Technology Graduate University, Okinawa 904-0495, Japan

**Keywords:** critical dynamics, collective behavior, phase transition, the Ising model

## Abstract

Social insects such as honey bees exhibit complex behavioral patterns, and their distributed behavioral coordination enables decision-making at the colony level. It has, therefore, been proposed that a high-level description of their collective behavior might share commonalities with the dynamics of neural processes in brains. Here, we investigated this proposal by focusing on the possibility that brains are poised at the edge of a critical phase transition and that such a state is enabling increased computational power and adaptability. We applied mathematical tools developed in computational neuroscience to a dataset of bee movement trajectories that were recorded within the hive during the course of many days. We found that certain characteristics of the activity of the bee hive system are consistent with the Ising model when it operates at a critical temperature, and that the system’s behavioral dynamics share features with the human brain in the resting state.

## 1. Introduction

Social insects are capable of creating and supporting sophisticated modes of spatial organization; furthermore, they possess the ability for decision-making at the colony level. What is the basis of this collective cognitive capacity? Several lines of research seek to elucidate what traits are shared by entities capable of such adaptive behavior, regardless of whether their mode of operation is individual or collective [1]. For the case of individual animals, a prominent conjecture conceived through the cross-pollination of statistical physics and experimental neuroscience is known as the “Critical brain hypothesis”. It holds that the remarkable adaptive abilities of neural systems require them to be poised in the vicinity of the critical state, because in this way, the system has access to the widest repertoire of dynamic patterns. This claim is supported by a notable amount of experimental data, recorded both in vitro and in vivo. At the same time, there is evidence that some collective entities such as swarms of midges and flocks of birds exhibit hallmarks of the critical state at the collective level [2,3,4]. However, a comprehensive assessment of criticality was not carried out for any of the major types of eusocial insects, such as ants, bees, and termites. Recent advances in scientific image acquisition and analysis techniques now allow for recording trajectories of individual insects [5]. In our work we analyzed a dataset of honey bee trajectories, focusing on the hallmarks of the critical state.

### 1.1. Smarts in Numbers—Collective Intelligence

Eusocial insects’ abilities in foraging [6], enacting complex spatial arrangements of their bodies [7,8] and creating elaborate nesting structures are well known. Furthermore, they can handle making decisions on a more abstract level. For example, a honey bee colony produces a daughter colony when the original site becomes overcrowded [9]. The choice of nest site depends on a number of parameters and an erroneous decision could lead to the inevitable demise of the colony. Thus, when swarming, bees aggregate themselves into a cluster, usually in a form of penchant hanging from a tree brunch. Then, the collective considers information brought by scout bees about the relative quality of the nearby locations. The swarm evaluates the merits of the proposed locations and makes the choice which is then followed through unanimously, in the overwhelming majority of cases. Everyday functioning of the colony also requires the handling of information to optimize foraging and allocate tasks for individual bees.

Ants display similarly complex behavior on the colony level. They are using cooperation to extend their sensing range [10] and engage in complex physical tasks, such as transporting heavy objects or building elaborate structures. Furthermore different colonies demonstrate distinct differences in behavioral patterns across several behavioral traits, thus having a degree of “collective personality”, which affects their evolutionary fitness [11].

At an abstract level of description, the mechanisms of collective decision-making share common features with the mechanisms of cognition within the brain [1]. Famous entomologist Thomas D. Seeley [9] noted in his book remarkable similarities between the way in which a honeybee swarm performs the task of making a decision for a new nest site and the activity of a primate brain during a perceptual discrimination task. In both cases, the correct decision is acquired through the non-linear aggregation of activity of individual constituents of the system. Other studies showed [12] that bee colonies adhere to the same psychophysical laws that humans do when making decisions while facing varied and conflicting sources of information.

A fundamental organizing principle shared by the brain and eusocial insects is that all forms of behavior and cognitive ability arise from the local interactions between the system’s elements, be it insects or neurons, and does not need to be explicitly organized with a blueprint or central pacemaker of some sort.

### 1.2. Critical Brain Theory

In the neuroscience field, it has been conjectured that healthy brains are poised in the vicinity of second-order phase transition [13,14,15,16]. Unlike the more familiar case of first-order transition, exemplified by the freezing of water, the second-order transition does not have a sharp boundary separating the two phases. In the vicinity of the critical point, both states of matter could coexist and the system displays a set of remarkable characteristics: most notably, long-range correlations link distant locations, and events of all scales could occur in the system [17].

Recordings of Local Field potentials (LFPs) from cortical slices [18], as well as an analysis of neural data of different modalities, including EEG, MEG, and fMRI [19,20], showed that both the isolated neural tissue and the intact functioning brains exhibit statistical properties characteristic of the system in the critical state. Most importantly, the size of fluctuations in the system scales abides by the power-law scaling, making events of all sizes possible, long-range correlations exist in both temporal and spatial domains, and activity patterns exhibit complexity and variability, significantly exceeding what would be expected by random chance. Furthermore, research in the field showed that such a state might be not epiphenomenal (i.e., consequent of but not causal to), but necessary to maintain the brain’s functionality [15].

This view, which became known as the “critical brain theory”, conjectures that the critical state of the system underlies its key functionality, including its ability to produce a wide range of emergent configurations of activity. Emergent behavior is paramount for an adaptive system, as it underlies its ability to produce a multitude of responses without altering the underlying structure.

It has been established that in the vicinity of the critical point, the system has the widest repertoire of dynamic patterns emerging from local interactions only [21]. A body of evidence, accumulated through computational models and experiments with cortical slices, shows that critical state neural networks are best suited for complex computation. Information transmission, integration [22], and representation [23] are optimal, and dynamic range is maximized [24], allowing for the networks to respond to a wide range of stimuli. These findings are corroborated by medical research showing that deviations from criticality are correlated with suboptimal mental states in humans [25].

### 1.3. Evidence for Criticality in Collective Behavior

Evidence of criticality and its implications discussed above in Section 1.3 applies mostly to mammalian brains. As we argue in Section 1.1, insect communities, despite having little structural resemblance to the brain, share the same burden of navigating the world and making choices crucial to survival.

The universality of physical laws implies that different systems could be described with the same set of basic laws. At the same time, principles of convergent evolution make it possible that similar phenomena could come to be in unrelated species if they are advantageous to survival. Given the benefits that the critical state confers to the adaptive ability of the systems, it is fitting to conjecture that swarms and colonies of collective insects would evolve to be positioned in its vicinity [4,26]. Investigations of brains’ critical phenomena focus on analyzing patterns of electric signals (EEG, LFP, MEG) or alternatively using proxy metrics for brain activity (fMRI). Animal collectives use many modes of interaction to exchange information and synchronize actions within the group, including vocal cues, movements, and phenomenal signals. Previous research we reviewed assumes, unanimously, that the spatial and temporal scale at which critical phenomena would presumably be exhibited by such systems makes the movement patterns, correlations, and derivatives thereof the best venue for analysis.

Recent research showed that critical phenomena are observable in flocking birds [2] and swarming of midges [3,27]. Long-range velocity correlations underlie the astonishing ability of these creatures to maneuver collectively with the synchronicity of one. Cursory evidence exists to suggest that ants make use of enhanced coordination between individuals conferred by criticality to maximize the load-carrying capacity of a group [28] and optimize forage route allocation [29].

## 2. Materials and Methods

### 2.1. Bee Data

All bee data analyzed in this paper were provided by Professor G. Robinson and his team. Their recent paper explains the nuances of the methods [5]. To summarize briefly: five recording sessions, henceforth refereed to as trials, were conducted, during each, a different colony of 1200 worker bees was recorded for several days and nights. Insects were marked with barcodes attached to the thorax. Colonies resided in the rectangular hives which had video cameras installed inside. Barcodes enabled tracking of the positions of individual insects. Several days and nights of activity were recorded for each colony. For the first two days and two nights of each trial, hives were kept sealed and supplied with the necessary nutrients. Afterward, bees were allowed to leave at will to scout and forage.

The raw dataset contained IDs and coordinates accompanied by UNIX timestamps. We developed a prepossessing pipeline using linear interpolation to re-sample individual trajectories at uniform time intervals. Data were partitioned into day and night intervals. Inactive bees, for which it was impossible to reconstruct the trajectory accurately, were filtered out. During our analysis, we focused on the solar days when bees were sealed inside the hive, assuming that such a condition is similar to the resting state of the brain.

For each insect, we compute its kinetic energy from the horizontal and vertical displacement over time (Equation 1). Such an approach was first implemented [30] to identify bursts of activity in bee hives. It should be noted that the purpose of computing kinetic energy is that it provides a convenient proxy to measure the change in activity over time, such that physically, more rigorous details such as accounting for the mass of each bee, could be omitted. Figure 1 illustrates example trajectories and their kinetic energy is computed. To characterize the activity of a hive as a whole, mean kinetic energy $K^{mean}$ is computed according to (Equation 2).
(1)$$K^{bee}\left(t\right)=\Delta x^{2}+\Delta y^{2}$$
(2)$$K^{mean}\left(t\right)=\sum_{i=1}^{n}K_{i}$$

### 2.2. The Ising Model

The Ising model was originally developed to describe ferromagnetism; however, it has been successfully implemented to model disparate phenomena, ranging from statistical physics to economics [31]. Its relevance to our research is that when the spatial dimensions of the model exceed 1, then the system could undergo phase transition and exhibit critical behavior (the one-dimensional Ising model could exhibit critical behavior, however, phase transition is impossible in the 1d system). In our work, we used a two-dimensional model on the toroidal grid. It consists of a square lattice with sides of length *L* composed of N=L×L sites with the nearest-neighbor interactions. Each site has an associated binary “spin” variable $s_{i}=\pm 1$. Lattice configuration is uniquely specified by a sequence of spin variables. For a system-level description of the model, its order parameter—Magnetisation (Equation 3), which corresponds to the mean spin of the model, is computed. The energy of different lattice configurations in the absence of an external magnetic field is given by the Hamiltonian function (Equation 4), where *J* represents interactions between different lattice sites.
(3)$$M=\frac{1}{N}\sum_{i=1}^{N}s_{i}$$
(4)$$E=-J\sum_{i,j=nn(i)}^{N}s_{i}s_{j}$$

Neighboring spins tend to align with each other, where the probability of alignment is controlled by the temperature parameter *T*. Competition between thermal fluctuations, which induce stochastic behavior, and the nearest-neighbor interactions, which pull the system towards a more ordered state, governs the dynamics of the model. At low temperatures $T=T_{low}$, the system soon finds itself in a quiescent state. High temperature $T=T_{high}$ causes uncorrelated, random activity. In the vicinity of the critical point, $T\approx T_{crit}$ the system exhibits a multitude of critical phenomena, such as long-range temporal and spatial correlations and complex patterns of activity.

We implemented the Metropolis Monte Carlo algorithm [32] to solve the equilibrium configuration of the model. Details of the algorithm are given in the Section A.1. Each step of the algorithm consists of *N* site updates. Initially, the system is allowed to settle into an equilibrium state. During this process, known as thermalization, no recordings are taken. In our implementation, this stage lasts 10,000 steps. Then, 2000 consecutive configurations of the model, separated by one step of the algorithm, are recorded. We used 3 Ising models of size N=10,000 with their *T* parameter set, respectively, at $T_{low}=2.0$, $T_{crit}=2.3$ and $T_{high}=3.0$.

It should be noted that the critical temperature for the Ising model in 2 dimensions has been computed analytically [33]; however, the exact solution $T_{crit}\approx 2.2691$ is only valid at the thermodynamic limit when the infinite lattice is considered. In numerical simulations with finite systems, an approximate critical temperature $T_{crit}=2.3$ is used.

We have tested models of different *N*, obtaining qualitatively similar dynamics. Statistical tools that we implemented to characterize the dynamics of the model dictate that a system with large *N*, as specified above, has to be used to obtain reliable estimates of parameters.

### 2.3. Dynamic Correlations

Our approach of constructing graphs of dynamic correlations to compare unrelated systems from a statistical standpoint was influenced by a seminal paper [19] that used such an approach to compare brain dynamics inferred from resting state fMRI recordings with the Ising model at different temperatures. Thus, we tried to keep methodological consistency whenever it was possible. It should be understood that no direct correspondence between the structure of the Ising model and that of the honeybee colony is implied. The Ising model was used in our research solely as a vehicle to simulate critical phenomena: we believe that the systems we compared are so disparate that under normal circumstances, no statistically significant similarities would be found in their dynamics. Consequently, when such similarities are evident, then they are indicative of the critical phenomena present in both systems, not of the structural similarity between them.

To construct correlation networks, we treated each site of the Ising model and each bee as a node, while edges were inferred from correlations between the activity of different nodes (i.e., either between the kinetic energy time-series of individual bees or between the time-series of spins of different lattice sites of the Ising Model). Correlations were computed using Pearson’s correlation *r* coefficient (Equation 5), which captures the degree of linear correlation between the two time-series. Nodes were connected with edges when correlation exceeds some predefined threshold *p*. It is noteworthy that *r* is high only if both time-series exhibit similar dynamics, fluctuating from their respective mean values in a coordinated manner. For a pair of quiescent or stochastic time-series, *r* would be similarly low.
(5)$$r\left(x,y\right)=\frac{E[\left(x-\mu_{x}\right)\left(y-\mu_{y}\right)]}{\sigma (x)\sigma (y)}$$

A meaningful way to compare networks that have very different provenance is to scan a range of *p* values for each of the networks at hand, while simultaneously computing networks’ average degree 〈k〉. Then, comparisons could be made between networks of similar 〈k〉. Figure 2 illustrates this approach for Ising networks at 3 different temperatures and for the network derived from the correlations between the kinetic energy time-series of bees during 1 day. The exemplifying case of 〈k〉≈60 is used.

After constructing correlation networks for different systems, we investigated their degree distributions and other network metrics. The crucial step was to assess if the degree distribution of the system follows power-law. It has been brought up by the statistical community [34] that often methods used to characterize power-law relationships do not stand up to mathematical scrutiny. Conclusions on the nature of distribution based on examining the shape of the distribution or fitting data points with linear regression are often erroneous. The most reliable approach is to use maximum-likelihood to estimate the parameters of the distribution [35] and compare candidate power-law to other common heavy-tailed distributions. We followed the aforementioned methodology in our research; some nuances and supplementary information are contained in the Section B.2.

### 2.4. Temporal Analysis

Correlation-based network analysis elucidates the degree of correlation between time-series of different nodes. However, previous research concerning both neurological recordings [36] and computational models [37] which are known to exhibit critical behavior showed that such time-series exhibit significant autocorrelations—patterns of activity at one timestep significantly influence activity at a lagged time interval. An established method to quantitatively characterize the degree of autocorrelation in the time-series is the Hurst exponent *H* [38].

It is a scalar metric that ranges from 0 to 1. For random walks and patterns of the Brownian motion, in which successive timesteps are completely independent, H≈0.5. Such processes are called memoryless. Values of *H* close to 1 imply significant positive autocorrelations in the data, while values of *H* between 0 and 0.5 imply a negative autocorrelation.

Several approaches to computing *H* all have different pros and cons. We adapted a simplified methodology initially developed for stock market analysis [39] and rigorously tested our implementation on synthetic data. We transformed the mean kinetic energy time-series $K^{mean}\left(t\right)$ into the time-series of cumulative deviations from the mean value $\mu_{k}$ using (Equation 6).
(6)$$K^{sum}\left(t\right)=\sum_{i=1}^{i=t}K_{i}^{mean}-\mu_{k}$$

Subsequently, for all elements of the transformed series, we computed variance between elements at time *t* and time t+τ for a sequence of lags τ.
(7)$$Var\left(\tau \right)=\langle {\left(K_{t+\tau }^{sum}-K_{t}^{sum}\right)}^{2}\rangle$$

For the Brownian motion and other memoryless processes, variance was linearly dependent on lag: Var(τ)∝τ. However, when autocorrelations were present, this relation acquired an anomalous exponent: $Var\left(\tau \right)\propto \tau^{H}$. We computed *H* by solving for the relation between log(Var(τ)) and log(τ).

## 3. Results

### 3.1. Model Graphs

Figure 3 depicts degree distributions of correlation networks created using the Ising model at 3 different temperatures as well as an illustrative degree distribution for the correlation network computed using the kinetic energy time-series of bees recorded during one solar day. Earlier work used an analogous approach to compare fMRI correlation networks with the Ising model at $T=T_{crit}$ [19].

Several methods have been used to analyze criticality in neural data. We chose to use the network approach for our initial test for a number of reasons. Firstly, unlike fMRI voxels or neurons, bees could move freely inside the hive. Thus, methods which rely on measuring spatially localized activity, for example, the local field potentials in different areas of the neural tissue, would be challenging to apply to the bee data. Using correlations between the kinetic energy time-series of individual bees allowed for us to circumvent this difficulty, while still generating results compatible with the Ising model.

From Figure 3, it is evident that the degree distribution of the Ising model at the critical temperature and degree distribution of the bee correlation matrix share a similar long-tailed structure, regardless of the mean degree chosen for comparison. In contrast, at T=low and $T=T_{high}$, the Ising model has a strikingly different degree distribution. We chose to display only one day’s worth of kinetic energy measurements, yet we analyzed multiple days from different trials and acquired similar degree distributions for all of them. This full set of results is presented in Appendix Figure A2.

Besides these striking visual similarities, several key statistical characteristics are shared by the Ising model and the bee hive in the vicinity of the critical point, but are notably absent when $T=T_{low}$ or $T=T_{High}$. As shown in Table 1, the Ising model at $T=T_{crit}$ is best described by a power-law distribution, while at low and high temperatures, the best fit for the data is a log-normal distribution. Different power-law exponents, as well as the different numbers of nodes, explain the somewhat different shape of degree distribution for the Ising model at $T=T_{crit}$ and the bee hive; however, general features of heavy-tailed distributions are evident in both cases.

### 3.2. Network Metrics

Networks can be characterized by a number of other metrics, which account not only for the number of edges per node but also for the intertwined structure of the connections. Table 2 presents the most common [40] metrics computed for the correlation networks of the Ising model and different bee hives recorded during the experiment. Thresholds are set for all networks to have similar mean degrees. Additional information about the metrics we used is given in Section A.2 and supplementary tables describing the same networks with different thresholds and different mean degrees are available in Section A.2.

In order to compute characteristics such as average path length *L*, it is necessary for the network to be fully connected. Bee correlation networks have a significant number of isolated nodes, although the majority of nodes always belong to an interconnected component of significant size, regardless of the threshold value. The Ising model always remains fully connected at $T=T_{low}$ and $T=T_{high}$. Interestingly, isolated sub-graphs begin to appear in the network at $T=T_{crit}$ when threshold *p* is set to a higher value.

We extracted the largest component from the networks and used it to compute the metrics. Thus, 〈k〉 can vary within reasonable limits. A common technique in the study of networks is to compare the network at hand with a random graph [40]. For each of the networks, we created an ensemble of equivalent random Erdős–Rényi graphs and computed their average characteristics as a reference.

It can be noted that the correlation networks of the bee hive and of the Ising model at the critical temperature have much higher average clustering *C* than both their random equivalents and the correlation networks of Ising model at low and high temperatures. The average path length *L* in these clustered networks is generally somewhat higher than in the random networks, but considerably lower than if the graph had been a regular lattice with high clustering. Such a graph structure, which combines high clustering characteristics of lattice-like networks with a relatively low average path length of random graphs is known as the small-world network [41].

### 3.3. Temporal Correlations

As noted earlier, the critical state manifests itself not only through spatial correlations but also in increased autocorrelations within the time-series. We computed the Hurst exponent *H* to assess the presence of autocorrelations. High values of *H* imply that the system which generated the time-series is in the vicinity of the critical state. The mean magnetization *M* time-series of the Ising model is characterized by a high value of *H* only when *T* is close to $T_{crit}$ [37]. A common technique is to compare the time-series generated by the system of study with another time-series generated by a computational model which is known to exhibit critical behavior. For example, such a technique was used to show that temporal patterns of blackouts in the USA exhibit characteristics of the critical state [42].

Figure 4 exhibits different Hurst exponents computed for the Ising model at different temperatures, and an exemplifying case of the Hurst exponent computed for the $K_{mean}$ of the bee hive during a single solar day. *H* computed for other days shows only minor differences and each *H* value is always significantly higher than 0.5. This is indicative of long-term correlations within the time-series. As a sanity check we also computed *H* for randomly permuted mean kinetic energy time-series and, as it would be expected of essentially random data, we found that H≈0.5. The Ising model only displays high *H* values at $T=T_{crit}$, which is consistent both with previous research on the model and with the established connection between autocorrelations and criticality [37,42].

## 4. Discussion

### 4.1. Key Finding and Their Implications

In our work, we analyzed a dataset of bee trajectories in a way that allowed us to compare empirical data to a well-studied model. Such an approach has been previously used (e.g., [19]) to show that key dynamical characteristics of the human brain at its resting state exhibit a notable resemblance to the Ising model when the latter is in the vicinity of the critical state. Long-range temporal autocorrelations are also considered a hallmark of criticality in the Ising model and the neural time-series [43]. With some minor differences, which are to be expected given the different nature of the data, our analysis of bee correlation networks yielded remarkably similar results, namely similarity to the Ising model at $T=T_{crit}$ and lack thereof at low and high temperatures.

First, our observations are confirmed by the rigorous statistical analysis of the degree distributions. Second, the analysis of the correlation networks showed that certain characteristics, most notably clustering and path length, are comparable between the network of bee correlations and critical state of the model, but are drastically different otherwise. Third, analysis of the mean parameters $K^{mean}$ and *M* showed that recordings of the average activity of both the bee hive and the Ising model at critical states are marked with considerable autocorrelations within the time-series.

These similarities in the dynamics of the systems become even more noteworthy when we consider that in structural terms, both the Ising model and the human brain are drastically different from the bee hive. In the context of our analyses, differences in spatial structure are particularly important. The two-dimensional Ising model possesses only nearest-neighbor interactions, while the human brain is characterized by intricate connection patterns on the macro and micro levels. Although correlation networks are obtained solely from dynamics, one would expect them to be strongly influenced by the underlying spatial structure. The bee hive is quite different because the nodes of its correlation matrix represent agents largely unhindered by constraints in movement and interaction. Despite these dissimilarities, functional constraints on behavior are shared by the systems in question to a degree that precludes the appearance of dynamical similarities by mere chance. We believe that they signal a shared systemic property—the critical state, whose influence on the systems’ dynamics goes beyond specific structural constraints.

One potential concern with our comparative analysis is the choice of running the Ising model on a 2D grid, thereby imposing a local interaction constraint on the system. Alternatively, we could have implemented the model on a complete graph, in which all lattice sites are connected. Intuitively, that might seem a better approximation of the bee hive, with highly mobile agents. Empirical research and modeling studies of flocks and schools have demonstrated that in such systems, an ubiquitous strategy of the individual agents is to align their behavior with that of the nearby neighbors [44,45,46]. Information, however, could propagate through the whole system seemingly unencumbered by the prevalence of local connections. Such behavior is a clear mark of critical state. We do not have definitive information that bees follow such an alignment pattern inside the hive, yet the remarkable ability of the 2D Ising model with only local interaction to emulate a high-level description of bee hive dynamics, implies that a similar mechanism could be in play.

### 4.2. Critical Brains and Critical Swarms

Our results provide evidence in support of what we could call a “critical colony” hypothesis. As we had summarized earlier in Section 1.2, being in a critical state provides benefits to a system’s cognitive ability, to an extent that it can be argued that such a state is not simply advantageous, but even necessary. At the same time, there are convincing arguments in Section 1.1 for the claim that social insects are endowed with a sort of collective cognitive capacity, which, despite their entirely different mode of organization, shares key similarities with the capacities of vertebrate brains. Thus, we suggest that the benefits that the critical state brings would be desirable characteristic for both of these systems.

Previous research showed that hallmarks of the critical state are observable in swarming midges [3] and flocking birds [2]. Moreover, previous research on the same bee dataset found that the time-series of the $K_{mean}$ of the bee hive exhibits bursts of activity, significantly exceeding the average level, which is interspersed with quiescent periods [30]. The distribution of waiting times between these bursts abides by the power-law distribution. The authors did not discuss criticality as such, but it is noteworthy that this kind of distribution is also found to be the case for “avalanches” in the Back–Wisenfield sandpile model, a famous model of self-organized criticality [47], and for the patterns of the neural activity recorded in the cortical slices of mice [18] in vitro.

One of the benefits of the critical state in neural networks is that it enables an easier spread and integration of information. Our findings show that the correlation graphs of bee activity exhibit distinct small-world structures and contain hubs—nodes with exceptionally high degrees. A similar organization emerges in the Ising model when $T=T_{crit}$. Such network properties are also known to enhance the integration of information and facilitate the spreading of signals. In the human brain, there is a similar structural organization of cortical wiring [48,49]. The bee hive and the critical Ising model lack such structural constraints, yet their functional connectivity is remarkably similar. Uncovering such network structures in the functional correlations of the bees suggests that this type of organization is not only ubiquitous in the brain but is fundamental to the functionality of any cognitive system.

It is worth highlighting important differences between our work and the corpus of research that concerns hallmarks of criticality in the swarm dynamics, as well as with the studies employing tools of network science to study eusocial insects [5,50]. The most notable dissimilarity with the latter is that most studies focused on social networks, constructed by observing species-specific means of communication, such as trophallaxis in bees and attenuation in ants. Such contacts, important as they are, only constitute a fraction of individuals’ total activity, and they arguably account only for a portion of the total informational exchange that takes place inside the colony. Significant correlations between the kinetic energy time-series of individuals and highly specific degree distributions of the correlation graph, in our view, reflect features of the social system underlying collective cognition in the hive, aided by its critical state.

Hallmarks of critical behavior in collective entities, when studied empirically, were investigated in freely-moving agents: flocking birds or swarming midges. Thus, it is possible to argue that observed features, such as long-range velocity correlations, were begotten by the spatial order the organisms maintained and are transient in nature.

The movement data we analyzed were acquired when bees were locked inside the hive and provided with sustenance. They were under no pressure to maintain a specific movement order, as they would have been during swarming, or to move at all, in fact. Given these conditions, it is plausible that the hallmarks of the critical state revealed by our analysis would be observed only if such a state is inherent to the system, much like it is to the human brain at its resting state. Indeed, brain resting-state networks have been proposed to reflect a process of “constant inner state of exploration” that optimizes the system for a given impending input, thus influencing perception and cognitive processing [51,52]. It is exciting to consider the possibility that the colonies of eusocial insects exhibit a similar process.

## 5. Conclusions and Future Work

Our findings indicate that the honey bee colony inside the hive is a critical system. However, further work is required to ascertain the generalized conjecture that colonies of eusocial insects are critical in support of collective cognition. It is crucial to analyze other datasets to confirm that hallmarks of the kind we observed are ubiquitous across bee species and to analyze other eusocial insects, such as ants and termites. Other computational models of the critical state, such as the Ising model on a complete graph or the XY model with continuous spin dynamics [17] could also be compared with empirical data. Moreover, criticality manifests itself on different levels. We focused on temporal and spatial correlations; however, other aspects, for example the behavior of the system during coarse-graining, should also be considered. Another interesting direction for future research is to explore how the critical state is related to the hives’ normal daytime activity, such as foraging and swarming, as well as how it is modulated by environmental factors, for example by the day–night cycle. If our hypothesis is on the right track, we should find that increased evidence of criticality in the hives’ resting state is associated with improved collective decision making, such as more efficient foraging behavior.

Comparing high-level descriptions of different systems could shed light on general dynamical patterns of cognitive systems that are imperceptible when the focus is too narrow. Analysis of system-level properties such as criticality is a particularly promising approach in this regard. However, much ink has been spilled by some of the best scientific minds in debating the validity of the critical brain hypothesis and the debate is still not settled [53]. Evidence from other fields, such as provided here, could tip the scales in this debate toward a more widespread acceptance.

## Figures and Tables

**Figure 1 entropy-24-01840-f001:**
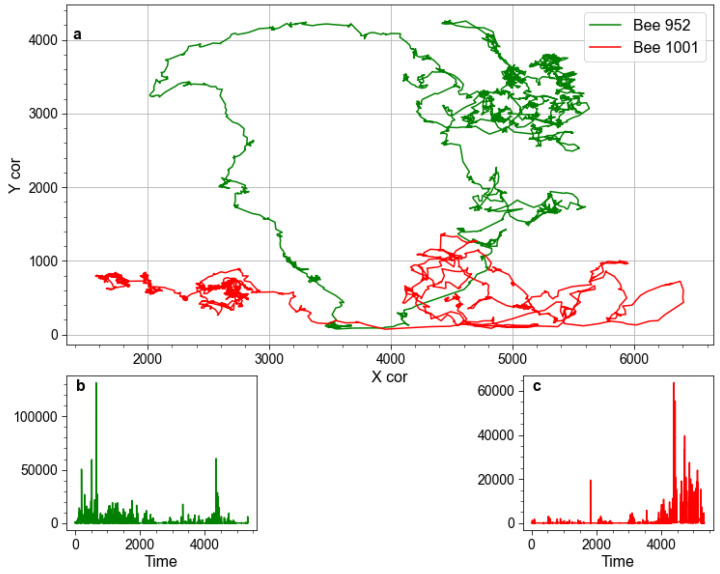
Illustrative example demonstrating how spatial coordinates are converted to kinetic energy. Pane (**a**) presents illustrative fragments of trajectories of two bees, each 5 s long, recorded during the first day of Trial 1. Panes (**b**,**c**), respectively, show kinetic energy time-series computed from their movements.

**Figure 2 entropy-24-01840-f002:**
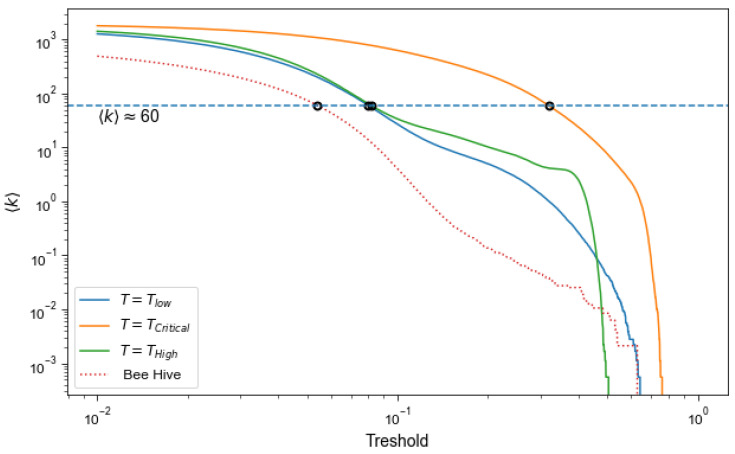
Dependence of the mean degree 〈k〉 on the threshold *p* used for binarisation. The dashed line shows a specific degree chosen to compare different networks, black circles indicate resultant relations between 〈k〉 and *p*.

**Figure 3 entropy-24-01840-f003:**
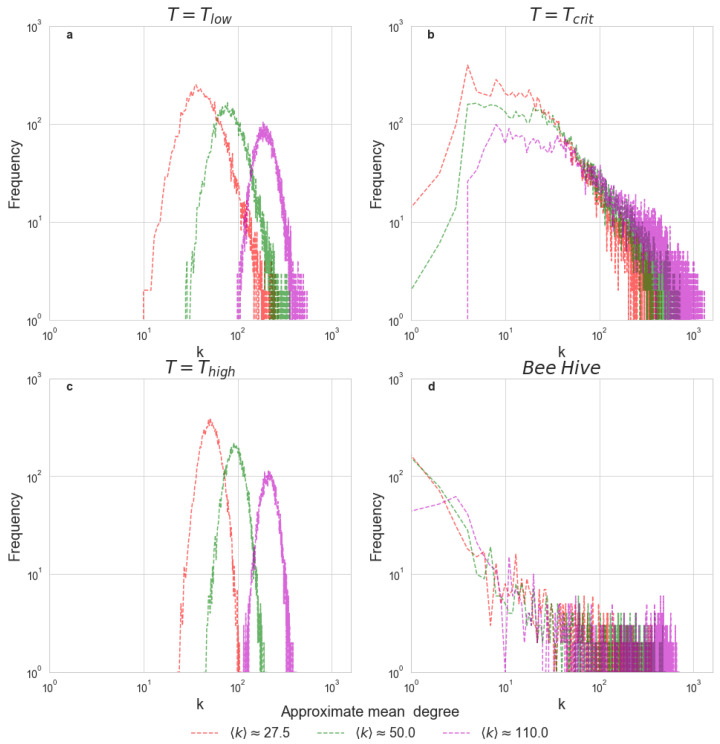
Degree distributions for the correlation networks. Each bin of the histogram corresponds to the frequency of nodes with a given degree. Panels (**a**–**c**) depict the degree distributions of the Ising networks at T = 2, T = 2.3 and T = 3. Panel (**d**) depicts an illustrative degree distribution computed using bee hive activity during 1 solar day, from sunrise to sunset. For each network, three representative values of 〈k〉 are plotted.

**Figure 4 entropy-24-01840-f004:**
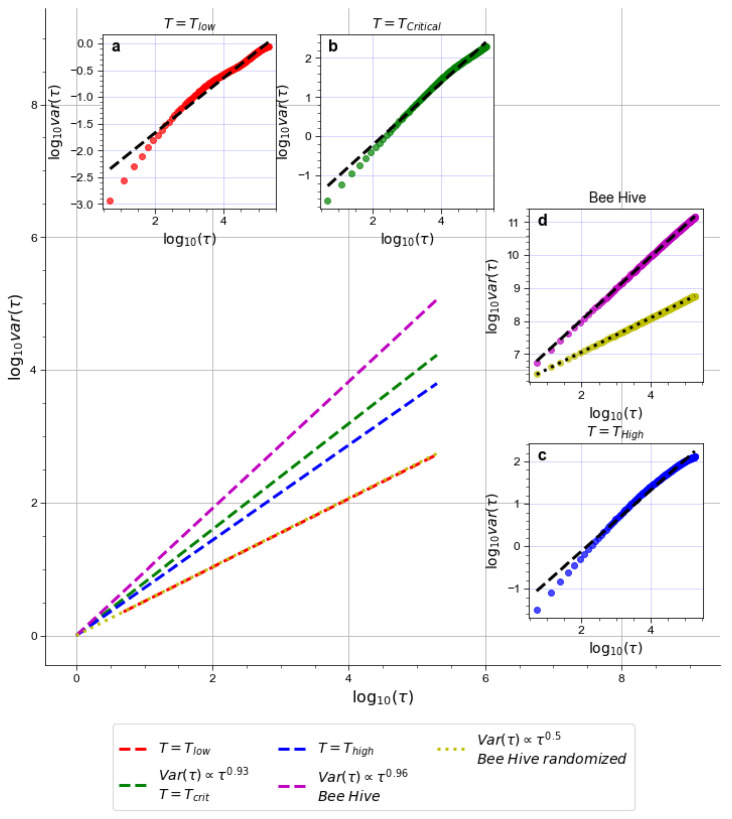
Hurst exponents for magnetization time-series of the Ising model and $K_{mean}$ time-series of the bee hive. The main pane shows the slopes of the Hurst exponents for different time-series in relation to each other. Inset panes (**a**–**d**) demonstrate how well *H* values fit the data. Note that although H values for $T_{low}$ and $T_{high}$ are considerably lower than for critical systems, their exact values are not given because they could be misleading for memory-less time-series.

**Table 1 entropy-24-01840-t001:** Characteristics of degree frequency distributions for various correlation networks. For log-normal distributions, the mean μ and standard deviation σ are given. Truncated power-law distributions are characterized by the power-law exponent α and $x_{min}$. The latter specifies the minimum *x* value of the data for which the power-law holds. All networks have a matching mean degree 〈k〉≈110.

System	Distribution Family	Parameters of the Distribution
Ising Model, $T=T_{low}$	*Log-normal*	μ=4.12, σ=0.132
Ising Model, $T=T_{crit}$	*Truncated power law*	α=1.99, $x_{min}=2$
Ising Model, $T=T_{high}$	*Log-normal*	μ=3.58, σ=0.225
Bee Hive	*Truncated power law*	α=3.49, $x_{min}=3$

**Table 2 entropy-24-01840-t002:** Network metrics computed for the giant components of correlation graphs of the Ising model (*T*) and bee hive (Trial). *N*—number of nodes, 〈k〉—mean degree, *C*—average clustering coefficient, *L*—average path length, *D*—diameter of the network. $C_{rand}$, $L_{rand}$, $D_{rand}$ refer to the metrics computed on ensemble of Erdős–Rényi graphs with *N* and 〈k〉 identical to the original network.

	*N*	〈k〉	*C*	*L*	*D*	$C_{\mathrm{rand}}$	$L_{\mathrm{rand}}$	$D_{\mathrm{rand}}$
$T=T_{high}$	10,000.0	111.51	0.09	2.04	3.0	0.02	1.98	3.0
$T=T_{crit}$	10,000.0	109.65	0.56	3.7	14.0	0.02	1.99	3.0
$T=T_{low}$	10,000.0	107.26	0.08	2.03	3.0	0.02	1.98	3.0
Trial5	1166.0	110.32	0.63	1.91	4.0	0.19	1.81	2.0
Trial4	1138.0	109.28	0.5	1.81	3.0	0.19	1.81	2.0
Trial3	1137.0	110.95	0.52	1.81	3.0	0.19	1.81	2.0
Trial2	1097.0	114.12	0.71	2.05	7.0	0.21	1.79	2.0
Trial1	946.0	111.74	0.61	1.8	4.0	0.24	1.76	2.0

## Data Availability

Data for this study were provided by the third party.

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
