# Peer review of "Evidence of Critical Dynamics in Movements of Bees inside a Hive"

_entropy, 2022, doi:10.3390/e24121840_

Round 1
Reviewer 1 Report
This is an interesting work, and gives a nice contribution to the hypothesis that collective behavior in some animal ensembles indicate criticality. I appreciated the two complementary views on this, one based on the correlation network structure and the other based on the fluctuation properties of some relevant quantity. It is a nice work, and I would recommend publishing it, after the authors clarify the following issues.
1. It is puzzling for me how the size of the simulated 2D Ising system was chosen. A very obvious choice would have been to match the number of spins in the Ising system with the number of bees. Taking into account that the system sizes are very different, and that for scale-free type graphs the average degree can vary with the size of the network it is not obvious at all why the average degree should be matched. Probably the authors should comment on this aspects.
2. The degree distributions presented in Figure 3 should be constructed by the log-binning method, to smoothen the tail. As it is presented in Figure 3d now, there is
no convincing evidence for the power-law tail. Log-binning the experimental results could help.
3. For determining the H exponents at T=T_low and T=T_high there is not a good enough scaling in Figures 4a and 4c. These results clearly suggest that there is no scaling, so the forced H exponents are misleading. Even if the authors do not give values for H in these cases, their point is strong enough for T=T_critic (where indeed there is a nice scaling).
4. In equation (7) modulus has no meaning, since it is square….
5. In section 3.2 in “network metrics” why comparisons are done with he giant component of Erdos-Renyi graphs? (Renyi misspelled…”Renui”) Of course it is possible to do this comparison, but what is the motivation for this?
6. Way to many misspelled words, notations and sentences. A very thorough spell-check and reconsideration of many sentences are necessary. The English should be improved. Just a few examples are below:
“A prominent conjecture conceived thought crosspolination of statistical physics and experimental neuroscience is known is the "Critical brain hypothesis"
“…consequent of but nor causal to …“
“tresholded”
“We extract we giant component from the networks and use it to compute metrics of equivalent random Erdos-Renui graphs”
“fully connected at T = Tlow and T=Thigh.”
“not only in through spatial correlations, but also in the increased autocorrelations within the time-series “
“at T = low and T = Thigh Ising model has a strikingly different degree distribution.”
For the Ising model in 2D misused “chaotic behavior” it should be “stochastic behavior”
Reviewer 2 Report
please find attached

Round 2
Reviewer 1 Report
Additional Minor comments (to be corrected before publication):
1. Section 2.2. Note that critical behavior is present also in the 1D Ising model when T->0....
2. In equation (5) what is the difference between X and x (or Y, and y),
it should be the same quantity? or x=X-_\mu_x.? Please state.
3. In the appendix A, at item 5 at point i). in the exponent a (-) sign is missing in front of \delta E.
Author Response
Authors thank the reviewer for more useful remarks.
1) We added a footnote about this.
2) They are the same quantity. Different letter size was misleading, we corrected it.
3) Corrected
Reviewer 2 Report
The Authors properly replied on my suggestions, I recommend the paper for publication. Just one small remark: in their reply the Authors say that they "have added this review along with another relevant recent opinion paper to
the introduction. It was a very useful summary of the field." However I do not see this review (i.e. a preprint arXiv 2211.03879) in the list of references. Does this mean that the current version of the paper is not the final one? To make the things clear: I am not the author of the above mentioned review.
Author Response
We thank the reviewer for useful and insightful comments.
Actually, we simply forgot to include the aforementioned archive review in the bibliography, although we have found it to be quite relevant.
We have corrected this along with a few minor typos and a missing minus sign in the latest version of the article.